# Automatic Recognition of Ragged Red Fibers in Muscle Biopsy from Patients with Mitochondrial Disorders

**DOI:** 10.3390/healthcare10030574

**Published:** 2022-03-19

**Authors:** Jacopo Baldacci, Marco Calderisi, Chiara Fiorillo, Filippo Maria Santorelli, Anna Rubegni

**Affiliations:** 1Kode Solutions, 56125 Pisa, Italy; j.baldacci@kode-solutions.net (J.B.); m.calderisi@kode-solutions.net (M.C.); 2Paediatric Neurology and Muscular Diseases Unit, University of Genoa and G. Gaslini Institute, 16147 Genova, Italy; chi.fiorillo@gmail.com; 3Molecular Medicine for Neurodegenerative and Neuromuscular Diseases Unit, IRCCS Stella Maris Foundation, 56128 Calambrone, Italy

**Keywords:** muscle biopsy, ragged red fibers, machine learning, image recognition, computer-aided diagnosis

## Abstract

Mitochondrial dysfunction is considered to be a major cause of primary mitochondrial myopathy in children and adults, as reduced mitochondrial respiration and morphological changes such as ragged red fibers (RRFs) are observed in muscle biopsies. However, it is also possible to hypothesize the role of mitochondrial dysfunction in aging muscle or in secondary mitochondrial dysfunctions. The recognition of true histological patterns of mitochondrial myopathy can avoid unnecessary genetic investigations. The aim of our study was to develop and validate machine-learning methods for RRF detection in light microscopy images of skeletal muscle tissue. We used image sets of 489 color images captured from representative areas of Gomori’s trichrome-stained tissue retrieved from light microscopy images at a 20× magnification. We compared the performance of random forest, gradient boosting machine, and support vector machine classifiers. Our results suggested that the advent of scanning technologies, combined with the development of machine-learning models for image classification, make neuromuscular disorders’ automated diagnostic systems a concrete possibility.

## 1. Introduction

Mitochondrial diseases represent a group of metabolic disorders with a common link of impaired mitochondrial function producing a chronic state of energy failure.

With a prevalence of 1/12,000 individuals, mitochondrial diseases have an extremely variable phenotype and can present at any age [1]. The most involved tissues are those with a high metabolic demand, such as the central nervous system, skeletal muscle, and heart.

Further complexity arises as a result of the dual genomic expression of mitochondrial proteins from both nuclear and mitochondrial DNAs [2].

A multidisciplinary approach to the diagnosis of mitochondrial disease is to integrate information from clinical, histochemical and biochemical tests in order to target molecular genetic screening [1]. Recently, the traditional diagnostic approach requiring histopathological investigations in muscle biopsy and the study of the oxidative phosphorylation (OxPhos) enzyme in muscle samples (“biopsy first”) has been replaced by massive gene testing-adopting methodologies of next-generation sequencing which target a few hundred genes or the whole exome (“genetics first”) [3,4,5,6]. The latter approach attempts to disentangle the associated clinical phenotypes based on the genotype (“reverse phenotyping”) [7] but often lacks the direct view of the consequences in disease tissues necessary to clarify uncertain cases [8].

In fact, many patients with mitochondrial disorders show histological and histochemical changes indicative of OxPhos dysfunction such as cytochrome *c* oxidase-negative fibers or increased Succinic dehydrogenase stain and the presence of so-called “ragged red fibers” (RRFs) with Gomori’s trichrome stain (Figure 1). RRFs represent the accumulation of abnormal mitochondria below the plasma membrane of the muscle fiber and are considered a sort of “hallmark” of mitochondrial dysfunction. Their tissue recognition might facilitate gene prioritization in primary mitochondrial disorders.

With the introduction of slide scanners in the histology work-flow, the microscopic examination of many kinds of tissue can now be performed on specialized software [9]. In cancer research, for example, the use of digital slides to examine disease-specific biological characteristics and quantitatively assess tumor progression is a common practice [10].

Moreover, the lack of uniformity in the diagnosis is another important issue in the management of mitochondrial diseases. Artificial intelligence techniques in digital slides could contribute to the achievement of agreed and shared criteria, which are often missing especially in the diagnosis of ultra-rare disorders.

The aim of this study was to develop and validate machine-learning methods to detect RRFs in digital light microscopy images of skeletal muscle sections. We designed and tested a classification system based on color, texture, and wavelet transform features able to predict the presence of RRFs in the images.

## 2. Materials and Methods

Histological images data were collected from 9 patients affected by mitochondrial disease (Appendix A). Each image contained at least one RRF. For light microscopy acquisitions, we used a Zeiss AxioCam MRc5 camera mounted on a Zeiss Imager.M2, and we used the acquisition software AxioVision SE64 rel 4.9.1. All images were acquired at a 20× magnification with a dimension of 1653 × 1239 pixels (format: JPEG; resolution: 96 DPI).

From each image acquired, we generated 165 sub-images with a size of 110 × 110 pixels (15 × 11 sub-images; Figure 1). We then collected all the sub-images that contained an RRF, and we manually annotated them as “ragged” (100 sub-images). After that, we arbitrarily collected a similar number of sub-images that did not contain RRFs, and we manually annotated them as “not ragged” (138 sub-images). Finally, we also arbitrarily collected some images manually annotated as “waste” (250 sub-mages). These sub-images included the areas of the image where:There were artifacts (bubbles, slice folding, staining artifacts, etc.) that did not allow a clear view;The percentage of the connective tissue was higher than in the muscle tissue.

With this method, we composed the dataset for machine-learning models, and it resulted in a total of 488 sub-images.

All the sub-images that comprised the dataset were used as inputs for generating features. The generated features were grouped into color, texture, and wavelet transform features.

Color features: the HSV (which stands for Hue, Saturation, Value) color model is a non-linear transformation from the RGB (red, green, blue) color space that can describe a perceptual color relationship more accurately. In this paper, the HSV color space was adopted for color feature extraction. All RGB color sub-images were transformed to HSV color sub-images. Thirteen first-order statistical features were extracted from each channel’s histogram, for a total of 39 color features.

Texture features: color sub-images were converted to grayscale images. A total of 43 second-order texture features were derived from three matrices:The grey-level co-occurrence matrix (GLCM) of size *Ng × Ng* is defined as *P(i,j|δ,θ).* The *(i,j)* element of this matrix represents the number of times. The combination of levels *i* and *j* occurs in two pixels in the image, which are separated by a distance of *δ* pixels along angle *θ* [11].Twenty-one second-order GLCM features were calculated.The grey-level run length matrix (GLRLM) quantifies gray-level runs, which are defined as the length in the number of pixels, of consecutive pixels that have the same gray-level value. In a gray-level run length matrix *P(i,j|θ)*, the *(i,j)th* element describes the number of runs with gray level *i* and length *j* occur in the image along angle *θ* [12].Eleven second-order GLRLM features were calculated.The grey-level size zone matrix (GLSZM) quantifies gray-level zones in an image. A gray level zone is defined as the number of connected pixels that share the same gray level intensity. A pixel is considered connected, if the distance is 1, according to the infinity norm (8-connected region in two dimensions). In the gray-level size zone matrix *P(i,j)*, the *(i,j)th* element equals the number of zones with gray level *i* and size *j* appears in the image. Contrary to the GLCM and the GLRLM, the GLSZM is rotation-independent, with only one matrix calculated for all directions in the image [13].Eleven second-order GLSZM features were calculated.

Wavelet transform features: the wavelet transform uses different basis functions to split images into multiple frequency scales described as high and low frequency details. In a two-dimensional (2D) wavelet, four quarter-sized images were generated through the combined action of low-pass and high-pass filters in each of the columns and rows of the image. The wavelet filter used in the processing of the wavelet transform is a least asymmetric Daubechies basis function with a length of 8. Thirteen first-order statistical features for each resulting 4 sub-images were computed.

The dataset was composed of 250 “waste” images, 100 “ragged” images and 138 “not ragged” images. A total of 134 features (39 color features, 43 texture features, and 52 wavelet transform features) were extracted from each sub-image. The features were normalized to a zero mean and unit variance.

We performed a double-step binary classification. The first step aimed to distinguish muscular tissue from the non-relevant zones of the images. For the first step, we used the whole dataset: the waste sub-images were labeled as “waste”, and the “ragged” and the “not ragged” sub-images were labeled as “tissue”. The second step aimed to distinguish RRFs from “healthy” muscle fibers. For the second step we used the “tissue” sub-images only: the ragged sub-images were labeled as “ragged”, and the “not ragged” sub-images were labeled as “not-ragged”.

In both steps, the dataset was split into a training set (80%) and a test set (20%) to train and test the machine-learning models. The training set contains a known output, and the model learns with these data in order to be generalized to new data.

Both classifiers were built choosing the best model between 3 supervised machine-learning algorithms:Random forest (RF): This is a widely used shallow machine-learning method that combines decision tree predictors following the bagging technique [14,15]. In this model, the class that receives the majority of votes from trees in the forest is considered the output result. This protocol relies on creating the number of models (n) and averaging predictions of all models for a finale prediction.Gradient boosting machine (GBM): This is an ensemble algorithm of decision trees [16,17]. The ensemble works by combining a set of weaker machine-learning algorithms to get an improved machine-learning algorithm in overall. The main difference between GBM and RF is the way of sampling. RF is based on uniform sampling with return. Instead, GBM gives higher weights to the wrongly predicted samples in the current weaker leaner, and then these samples will be paid more attention to when training the next weaker leaner.Support vector machine (SVM): This system divides the training set into two parts by constructing a hyper-plane in the feature space. Features in non-linear separation may be changed into linear separation, using kernel functions to map the original data to a feature space with a higher dimension [18,19].

In an attempt to prevent overfitting, we used 10-fold cross-validation to train classifiers. The training dataset was randomly divided into 10 roughly equally numbered, non-overlapping subsets, each called “a fold”. Then, nine folds were used as the training set, and the last one as the validation set. Using each of the 10 folds as the validation set, the above process was repeated 10 times.

The generations of each model for predictions required tuning of key hyperparameters. The value of each parameter was chosen by using random search at each step of the cross-validation.

For each model, the sensitivity (SEN), specificity (SPE), false positive rate (FPR), false negative rate (FNR), positive predicted value (PPV), negative predicted value (NPV), accuracy (ACC), and F1 measure (F1) were calculated to choose the best hyperparameters values. After retraining, the performance of the tuned models was further assessed on the test set by constructing the corresponding receiver-operating characteristics (ROCs). The ROC curve illustrates the tradeoff between the SEN and the SPE as a cut-off point for decision making. Finally, the area under the ROC curve (AUC) was estimated for each model. Appendix A shows a summary outline that well explains the entire process.

All the aforementioned methods were performed using the programming language. R. *radiomics* package was used to extract color features and texture features. *waveslim* package was used to perform wavelet transforms on the images. Machine-learning models were built using the package *mlr.* ROC curve calculations were performed using the package *ROSE.*

## 3. Results

We designed an experimental setting to demonstrate automated learners’ ability to identify the following:The wasted zones in Gomori’s trichrome-stained muscle images;RRFs in a field of normal muscular fibers.

We benchmarked 3 machine-learning models in order to select the one with the best overall fit for both classification steps. Eight classification metrics were estimated using 10-fold cross-validation for the training set. Table 1 shows performance metrics. Each model was fine-tuned optimizing the hyperparameters in order to achieve the highest scores for each model (priority to F1 metrics). Such models were used to estimate the ROC curve and the AUC of the test set.

### 3.1. Waste-Tissue Classification

The F1 scores for RF, GBM, and SVM for the waste-tissue classification step were 0.87, 0.91, and 0.89, respectively. GBM exhibited the best F1 score. In terms of other metrics, the accuracies of RF, GBM, and SVM were 0.87, 0.91, and 0.90, respectively (Table 1a). The tuned values of the hyperparameters for each model are shown in Appendix A. For each tuned and retrained model, the ROC curve and the AUC value were calculated on the test set. Figure 2a shows ROC curves. The AUCs of the tuned RF, GBM, and SVM were 0.89, 0.95, and 0.90, respectively. The tuned GBM model exhibited the higher AUC value of the test set, and the performance was significantly better than those of the rest models. We selected this model as the best waste-tissue classifier. Table 2a shows the confusion matrix of the test set.

### 3.2. Ragged–Not Ragged Classification

The F1 scores for RF, GBM, and SVM for the ragged–not ragged classification step are 0.96, 0.95, and 0.95, respectively. RF exhibited the best F1 measure score. In terms of other metrics, the accuracies of RF, GBM, and SVM were 0.96, 0.95, and 0.94, respectively (Table 1b). Appendix A indicates the tuned values of the hyperparameters for each model. For each tuned and retrained model, the ROC curve and the AUC value were calculated on the test set. The ROC curve for each model is shown in Figure 2b. The AUC values for ragged–not ragged graphs are ≥0.93. This indicated that RF, GBM, and SVM were able to separate ragged from “not ragged” fibers with a high confidence. The tuned RF model exhibited the higher AUC value of the test set. We selected this model as the best ragged–not ragged classifier. The confusion matrix of the test set is shown in Table 2b.

## 4. Discussion

Taken together, in our system, GBM and RF were selected as the best performing algorithms for the first and the second steps, respectively. Through the aforementioned model, we were able to obtain AUC values of >0.95 for both classification steps, and it is therefore possible to recognize, with a good reliability, the presence of RRFs on a tile that was generated from a 20× magnification image of Gomori’s trichrome-stained muscle tissue. We are aware that our sample size was relatively small for a machine learning analysis. The application of machine-learning models in rare diseases data is challenging because of the limited availability of training datasets.

Machine learning has been implemented in many areas of medicine with different uses in clinical practice [20,21]. Automated learning algorithms have been successfully evaluated in several types of cancer [22,23,24] and in microbiology [25], adopting pathological studies of tissues. As regards the muscle tissue, automated learning algorithms have been successfully evaluated in immunofluorescence images in several studies [26,27], and there are very few papers regarding histological images [28,29]. Each of these reports utilized algorithms able to recognize and segment cellular and environmental details and achieved good accuracy and performance. It must be said that convolutional neural networks (CNNs) were used as automated classification algorithms instead of machine-learning models in the majority of these works. CNNs remain the state of the art for image classification tasks, although automated learning is also used in biomedical images [30].

There are innovative elements in this paper. First, we have presented the first proof-of-concept study, where a mitochondrial histological “hallmark” of the disease can be automatically assessed in digital images. Beside RRF mitochondrial myopathies may present other pathological patterns such as cytochrome *c* oxidase-negative fiber, and this approach could be used to detect many morphological features in digital light microscopy images. Second, we have generated digital histology properties based on color, texture, and wavelet transform features from the input sub-images, in order to discriminate three subcategories to improve our definition of an RRF. Last, we have defined GBM as the best machine-learning model to achieve the classification of the wasted zones in Gomori’s trichrome-stained muscle images. In addition, the same machine-learning models can differentiate more properly RRF from normal fibers.

## 5. Conclusions

The goal of this paper was to take the first step in the automated recognition of histopathological patterns in muscle images in patients affected by mitochondrial diseases. We defined GBM as the best performing model for the waste-tissue classification step and RF as the best performing model for the ragged–not ragged classification step. The accuracies and the F1 measures performed by the three models (RF, GBM, and SVM) are comparable and very high, even if they adopted different approaches, further validating our results.

We are well aware that the fact that we did not use CNNs is a limitation of our article. In perspective, we should find the best CNN architecture for our classification tasks. This could produce an even better accuracy.

In order to set up a computer-aided diagnostic system in neuromuscular disorders, future works will require an automated recognition of other histopathological patterns beside RRFs. As a future work, we are going to try the application of this protocol in other additional morphological features of mitochondrial diseases and also in patterns of other neuromuscular disorders. Recent technology advances allow digital histopathology by scanning stained microscopic slides through the use of a commercially available scanner [31]. This automated system converts a glass slide to a digital whole slide image (WSI) preserving image resolution up to a 40× magnification. Although each WSI represents “big data”, interpretation is now made feasible using new image processing algorithms [32]. The advent of scanning technologies, combined with the development of machine-learning models for image classification, makes neuromuscular disorders’ automated diagnostic systems a concrete possibility.

## Figures and Tables

**Figure 1 healthcare-10-00574-f001:**
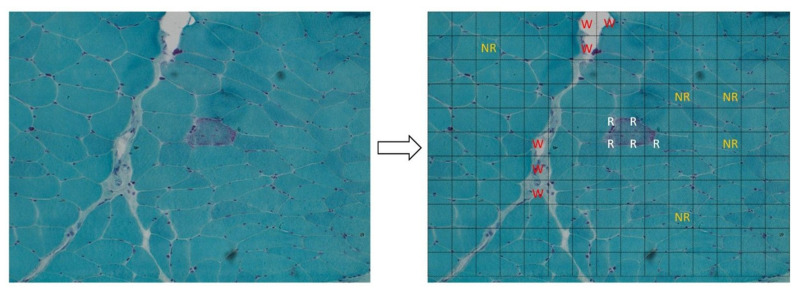
Left panel: ragged red fiber (RRF), hallmark of mitochondrial disorders, in Gomori’s trichrome stain (magnification: 20×; image acquired from the muscle of patient 8). Right panel: it shows the method used to create the dataset for our analysis: each acquired image was subdivided in 165 sub-images (15 × 11). All the sub-images that contain a portion of an RRF were added to the dataset and labeled as “ragged” (R label in the figure, written in white). After that, we arbitrarily collected a similar number of sub-images that do not contain RRFs, labeling them as “not ragged” (NR label, written in orange) and some “waste” sub-images (W label, written in red). The rest of the sub-images were discarded to avoid the unbalancing of the dataset.

**Figure 2 healthcare-10-00574-f002:**
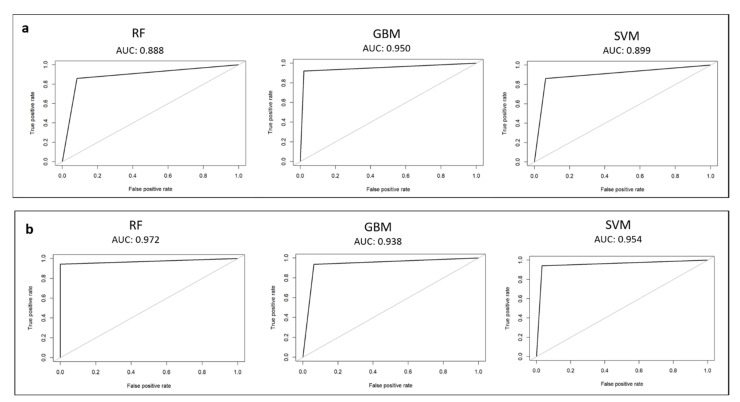
Receiver-operating characteristic (ROC) curve calculated on the test set for each model: (**a**) waste-tissue classification results; (**b**) ragged–not ragged classification results.

**Table 1 healthcare-10-00574-t001:** Performance metrics of each model: (**a**) waste-tissue classification results; (**b**) ragged–not ragged classification results.

a	F1 measure	Accuracy	True positive rate	True negative rate	False positive rate	False negative rate	Positive predicted values	Negative predicted values
**Random forest (RF)**	0.872	0.872	0.869	0.873	0.127	0.131	0.882	0.862
**Gradient boosting machine (GBM)**	0.908	0.911	0.892	0.924	0.076	0.108	0.929	0.895
**Support vector machine (SVM)**	0.891	0.900	0.883	0.922	0.078	0.117	0.920	0.887
**b**	**F1 measure**	**Accuracy**	**True positive rate**	**True negative rate**	**False positive rate**	**False negative rate**	**Positive predicted values**	**Negative predicted values**
**RF**	0.965	0.963	0.943	0.990	0.010	0.057	0.989	0.928
**GBM**	0.951	0.953	0.941	0.971	0.029	0.059	0.969	0.928
**SVM**	0.949	0.942	0.972	0.909	0.091	0.028	0.930	0.963

**Table 2 healthcare-10-00574-t002:** Confusion matrix of the test set for each model: (**a**) waste-tissue classification results; (**b**) ragged–not-ragged classification results.

**a**	**RF Confusion Matrix**			**GBM Confusion Matrix**			**SVM Confusion Matrix**		
		**waste**	**tissue**		**Waste**	**Tissue**		**Waste**	**Tissue**
	**Waste**	44	4	**Waste**	47	1	**Waste**	44	3
	**Tissue**	7	43	**Tissue**	4	46	**Tissue**	7	44
**b**	**RF Confusion Matrix**			**GBM Confusion Matrix**			**SVM Confusion Matrix**		
		**Not ragged**	**Ragged**		**Not Ragged**	**Ragged**		**Not ragged**	**Ragged**
	**Not Ragged**	30	0	**Not Ragged**	30	2	**Not Ragged**	30	1
	**Ragged**	1	17	**Ragged**	1	15	**Ragged**	1	16

## Data Availability

Not applicable.

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
