# Peer review of "Automatic Recognition of Ragged Red Fibers in Muscle Biopsy from Patients with Mitochondrial Disorders"

_healthcare, 2022, doi:10.3390/healthcare10030574_

Round 1

Reviewer 1 Report

The manuscript is not so easy to read at it is a very technical work going through machine learning-based classification/detection of diseases, here mitochondrial diseases. The content is interesting, indeed, this is a first step to machine learning-based technology in Medicine.

However, the manuscript needs to be improved to clarify and to help the readers, otherwise the manuscript will be to restrictive in the readers. I suggest to add a schema positioning the proposed machine learning-based model compared with the different machine learning-based models currently developped in the world. And so to clarify advantages and limits of the proposed model. It would be interesting also to introduce what could be next step to improve the proposed model.

Author Response

We thank the expert referees for their comments. A revised manuscript has been prepared with modifications considering all the comments of the expert reviewer (track changes in the revised text). A point-by-point response to reviewers has also been prepared.
The text has been edited for spelling and typos.
The references have been re-checked, and renumbered if necessary.

The machine learning models used in this study are well known in medicine application.
As you suggested, we added some articles about these models applied in medicine to our bibliography.
As we say in the discussion, convolutional neural network is the state of the art for image classification task. As a future work, we are going to apply this technique to our method and try to improve the results obtained in our classification tasks.

Reviewer 2 Report

In their work, Baldacci et al. developed and evaluated machine-learning methods for detection of ragged red fibers in muscle biopsy samples from patients with different mitochondrial myopathies. They compared three different supervised machine learning algorithms and identified the best methods for the two steps necessary to differentiate between: (i) waste vs. tissue, and (ii) ragged vs. not-ragged fibers. The study is very clear and innovative.

However, ragged red fibers can occur also in other myopathies than mitochondrial myopathies and in addition, mitochondrial myopathies without ragged red fibers exist. The authors should discuss these points more detailed. Their goal to set-up a computer-aided diagnostic tool thus requires additional morphological features to be taken into account.

Minor points:

Page 5, line 210 should read “We selected this model as the best ragged-not ragged classifier”.

Author Response

We thank the expert referees for their comments. A revised manuscript has been prepared with modifications considering all the comments of the expert reviewer (track changes in the revised text). A point-by-point response to reviewers has also been prepared.
The text has been edited for spelling and typos.
The references have been re-checked, and renumbered if necessary.

As suggested we added a sentence in the discussion section about different histological pattern in mitochondrial disease.
Moreover, we change the sentence Page 5, line 210 in "ragged-not ragged".

Reviewer 3 Report

In the manuscript titled “Automatic recognition of ragged red fibers in muscle biopsy 2 from patients with mitochondrial disorders”, the authors developed and validated a machine-learning method for ragged red fibers (RRFs) detection in light microscope images of the skeletal muscle tissue. They used image sets of 489 color images captured from representative areas of Gomori’s trichrome stained tissue retrieved from light microscope images at 20x magnifications. They compared the performance of random forest, gradient boosting machine and support vector machine classifiers. Their results suggest that the advent of the scanning technologies, combined with the development of machine learning models for image classification, make neuromuscular disorders’ automated diagnostic systems a concrete possibility. The manuscript is well-written. This proof-of-concept study indicated that a mitochondrial histological “hallmark” of the disease can be automatically assessed in digital images. Major comments: 1) Most figures are necessary to fully understand the entire process or protocol. 2) Figure 2 and Figure 4 should be presented as a table. 3) Ideally, the authors should try to apply this AI based diagnosis to at least one more disease. 4) Discussion should include study limitation (in addition to small sample size), and the potential application of this protocol for diagnosis of other diseases.

Author Response

We thank the expert referees for their comments. A revised manuscript has been prepared with modifications considering all the comments of the expert reviewer (track changes in the revised text). A point-by-point response to reviewers has also been prepared.
The text has been edited for spelling and typos.
The references have been re-checked, and renumbered if necessary.

We prepared a summary outline that well explains the entire process (supplementary fig. S1). Then, as you suggested, figure 2 and 4 are now presented as a table. Moreover, one of the study limitations is that we didn’t use the convolutional neural network as model for our tasks. In this paper we focused on the recognition of ragged red fibers as a first step in the digital pathology of mitochondrial disorders. As a future work, we are going to try the application of this protocol to other pathological patterns of mitochondrial diseases and also to patterns of other neuromuscular disorders. (Also using images of other muscle biopsy’s stains).
We added all the aforementioned points to our paper.

Round 2

Reviewer 1 Report

I thank the authors for the revision.

I do not have further comments.

Reviewer 3 Report

I have no further comments.